# Transformer Memory as a Differentiable Search Index

**Yi Tay**[*], **Vinh Q. Tran**[*], **Mostafa Dehghani, Jianmo Ni, Dara Bahri, Harsh Mehta**
**Zhen Qin, Kai Hui, Zhe Zhao, Jai Gupta, Tal Schuster**
**William W. Cohen, Donald Metzler**
Google Research
{yitay,vqtran,metzler}@google.com

## Abstract

In this paper, we demonstrate that information retrieval can be accomplished with a single Transformer, in which all information about the corpus is encoded in the parameters of the model. To this end, we introduce the Differentiable Search Index (DSI), a new paradigm that learns a text-to-text model that maps string queries directly to relevant docids; in other words, a DSI model answers queries directly using only its parameters, dramatically simplifying the whole retrieval process. We study variations in how documents and their identifiers are represented, variations in training procedures, and the interplay between models and corpus sizes. Experiments demonstrate that given appropriate design choices, DSI significantly outperforms strong baselines such as dual encoder models. Moreover, DSI demonstrates strong generalization capabilities, outperforming a BM25 baseline in a zero-shot setup.

## 1   Introduction

Information retrieval (IR) systems map a user query $q \in \mathcal{Q}$ to a ranked list of relevant documents $\{d_1, \ldots, d_n\} \subseteq \mathcal{D}$, typically represented by integers or short strings called *document identifiers* (docids). The most widely used IR approaches are based on pipelined *retrieve-then-rank* strategies. For retrieval, approaches based on inverted indexes or nearest neighbor search are common where contrastive learning based dual encoders (DEs) (Gillick et al., 2018; Karpukhin et al., 2020; Ni et al., 2021) are the present state-of-the-art.

This paper proposes an alternative architecture, wherein a sequence-to-sequence (seq2seq) learning system (Sutskever et al., 2014) is used to *directly* map a query $q$ to a relevant docid $j \in \mathcal{Y}$. This proposal is shown in the bottom half of Figure 1, for a sequence-to-sequence encoder-decoder architecture.

We call this proposed architecture a *differentiable search index* (DSI), and implement it with a large pre-trained Transformer (Vaswani et al., 2017) model, building on the recent success of large generative language models (LMs) (Brown et al., 2020; Raffel et al., 2019; Devlin et al., 2018; Thoppilan et al., 2022; Du et al., 2021). In this proposed architecture, all information of the corpus is encoded within the parameters of the Transformer language model.

At inference time, the trained model takes as input a text query $q$ and outputs a docid $j$. If desired, beam search can be used to produce a ranked list of potentially-relevant docids. As we show, this process can work surprisingly well when trained properly. In our experiments it can consistently outperform DE baselines, sometimes drastically: for a base-sized T5 model, Hits@1 on the smallest corpus is improved by more than 20 points, from 12.4% for a DE to 33.9% for DSI; and on a corpus

---

[*]Co-leads.

36th Conference on Neural Information Processing Systems (NeurIPS 2022), New Orleans, LA, USA.

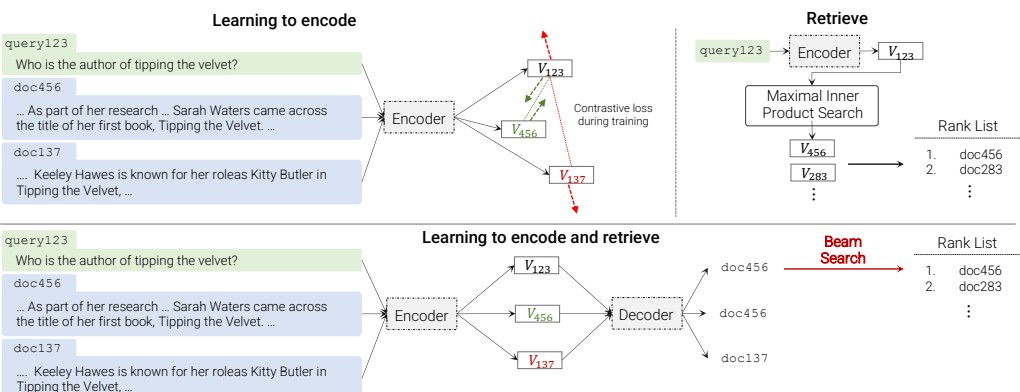

Figure 1: Comparison of dual encoders (top) to differentiable search index (bottom).

Table 1: Information retrieval requires a series of decisions, associated with the subproblems of document representation, indexing, and retrieval. Structured-document variants of DSI are also sensitive to a fourth decision, namely how docids are represented.

|  | BM25 or TFIDF | Dual Encoder (DE) | Differentiable Search Index (DSI) |
|---|---|---|---|
| doc/query rep. | sparse $\mathbf{v}_{d_j}$ vector in $\mathbb{R}^{|V|}$ | dense $\mathbf{v}_{d_j}$ vector in $\mathbb{R}^d$ | Various (see Section 3.1.2) |
| docid rep. | – | – | Various (see Section 3.2) |
| indexing | build inverted index mapping each term $t \rightarrow \{d_{j_1}, \ldots, d_{j_k}\}$ | build table mapping each docvec $\mathbf{v}_{d_j} \rightarrow j$ | train model (see Section 3.1.1) to map $d_j \rightarrow j$ |
| retrieval (top-1) | approximate sparse matmul to find $\text{argmax}_j \mathbf{v}_q^T \mathbf{v}_{d_j}$ | approximate MIPS to find $\text{argmax}_j \mathbf{v}_q^T \mathbf{v}_{d_j}$ | run trained model to find $\text{argmax}_j \Pr(j|q)$ |

$30\times$ larger, performance is improved by nearly 7 points. These gains increase when larger models are used: for an 11B-parameter T5 model, Hits@1 performance improves by more than 25 points over DE on the small corpus, and more than 15 points on the large corpus. DSI also performs extremely well in a zero-shot setting, e.g., improving Hits@1 by 14 points over BM25.

In addition to these quantitative gains, the DSI architecture is much simpler than a DE (see Table 1). A DE system fixes a search procedure (MIPS) and learns internal representations that optimize performance for that search procedure; in contrast, a DSI system contains no special-purpose fixed search procedure, instead using standard model inference to map from encodings to docids.

Of particular interest to the machine learning community, as Table 1 shows, in DSI *all* aspects of retrieval are mapped into well-understood ML tasks. This may lead to new potential approaches to solving long-standing IR problems. As one example, since indexing is now a special case of model training, incrementally updating an index becomes a special case of model updating (Sun et al., 2020).

In this paper, DSI is applied to moderate-sized corpora (from 10k to 320k documents), all of which are derived from one challenging retrieval task, and we leave the important question of the scaling DSI to larger corpora to future work. The task considered is retrieving supporting passages given questions from the Natural Questions (NQ) dataset, a challenging task for lexical models.

While the idea of DSI is simple, there are a number of ways it can be realized, some of which work surprisingly well, and some of which work surprisingly poorly. Below we explore a number of variations of the DSI architecture.

*Document representation.* We explore several approaches to representing documents, including a "naive" approach of using the document's full text, as well as variants of the bag-of-words representation used by traditional IR engines.

*Docid representation.* We look at several ways to represent docids. In addition to naively representing integers as text strings, we also consider *unstructured atomic docids*, where each document is assigned a unique token, and some simple baselines for constructing *structured semantic docids* that describe

how to navigate to a document through a hierarchical clustering of the corpus. Structured docids—either semantically structured via clustering, or *naively structured* as tokenized integers—scale better to large corpora, since the size of the vocabulary used in the decoder is made larger.

*Indexing.* A trainable IR system traditionally has two phases: indexing a corpus (i.e., memorizing information about each document), and learning how to effectively retrieve from the index. In DSI, the index is stored in the model parameters, and indexing is simply another kind of model training. Figure 1 suggests one approach to indexing a corpus: namely, to train on (1) examples $(d_j,\ j)$ that pair document $d_j$ with its docid $j$, in addition to (2) examples $(q,\ j)$ that pair a query $q$ with a relevant docid $j$. In this setup the examples of type (1) are "indexing" examples.

While it is clear that examples of type (2) alone do not provide enough information for a system to generalize to novel retrievals, there are many alternatives to examples of type (1) that might plausibly "teach" a model about the associations between documents and docids. We explore a number of these below, and show that some plausible-seeming techniques perform very poorly. We also explore a number of alternative multi-task optimization and curriculum learning schemes for combining these types of examples.

*Effects of model and corpus size.* Since recent results suggest that some properties of large LMs emerge only for very large model sizes Brown et al. (2020), we explore the performance of DSI for a range of model sizes and corpus sizes of 10k, 100k, and 320k documents.

*Summary.* We show that even naive representations for documents and docids, coupled with appropriate training procedures to fine-tune modern large LMs, can perform surprisingly well; we present two improved docid representations, unstructured docids and semantically-structured docids, which improve the naive representation choice. We show that there is substantial variation in performance among indexing/training strategies and we show that performance of DSI significantly and consistently improves with model scale. To our knowledge this is the first case of generative indexing improving performance over strong baselines for a well-studied document retrieval task.

## 2 Related Work

De Cao et al. (2020) describe a related sequence-to-sequence system called *autoregressive entity linking*, in which documents mentioning an entity—perhaps implicitly, e.g., by posing a question to which that entity is an answer—are mapped to a canonical name of that entity. In the case of Wikipedia, canonical entity names correspond to page titles, so this could be viewed as a sort of document retrieval. This approach has been adapted to other purposes, such as generating knowledge base triples in canonical form (Josifoski et al., 2021). The task we consider is different from those considered in autoregressive entity linking: our goal is to retrieve a document containing the answer, rather than a document whose title is the answer. More importantly, in autoregressive entity linking the generation target is a *semantically meaningful name*, whereas we allow targets to be *arbitrary docids*. This makes our approach applicable to general retrieval tasks, but raises new questions about docid representation and indexing strategies.

In autoregressive entity linking, generation is constrained to return an output from a fixed set. It would be feasible to constrain DSI generation outputs to be valid docids. Although we do not use this technique, the degree to which this might improve performance is a worthwhile question.

There is a large body of work on *retrieval augmented generation*, i.e., retrieving auxiliary documents to enhance language models (Borgeaud et al., 2021; Guu et al., 2020). These techniques are useful for many tasks including question-answering, but rely on traditional retrieval methods such as DEs. Here we use generation to *replace* a retrieval process, rather than using retrieval to augment a generation process.

Dual encoders (Dehghani et al., 2017; Gillick et al., 2018; Gao et al., 2021; Ni et al., 2021; Karpukhin et al., 2020) are a well-established paradigm for retrieval. The key idea is produce query and document embeddings independently and perform a similarity retrieval in vector space across all embedding pairs. Query and candidate documents are produced by a sequence encoder and training is performed using a form of contrastive loss.

The interpretation of a large Transformer model as a memory store have been investigated in prior work. (Roberts et al., 2020) demonstrated success on a closed-book QA task whereby they train

T5 models to retrieve facts that are encoded within the parameters of the model during pretraining. However, different from CBQA, the presented problem here in this paper is to retrieve full documents based on *docids* instead of generating direct answers. Meanwhile, (Petroni et al., 2019) also investigated language models as knowledge bases and found that pretrained LMs may already contain relational knowledge. (Geva et al., 2020) analyzes the knowledge encoded within Transformer feedforward layers. There have been also works that demonstrate the relation of Transformers to associative memory and Hopfield networks (Ramsauer et al., 2020), which reinforce the notion that Transformers should intuitively serve as a good associative memory store or search index.

# 3 Differentiable Search Index

The core idea behind the proposed Differentiable Search Index (DSI) is to fully parameterize traditionally multi-stage retrieve-then-rank pipelines within a single neural model. To do so, DSI models must support two basic modes of operation:

- **Indexing**: a DSI model should learn to associate the content of each document $d_j$ with its corresponding docid $j$. This paper utilizes a straightforward sequence-to-sequence (seq2seq) approach that takes document tokens as input and generates identifiers as output.

- **Retrieval**: Given an input query, a DSI model should return a ranked list of candidate docids. Here, this is achieved with autoregressive generation.

Following these two operations, a DSI model can be trained to index a corpus of documents and optionally fine-tune on an available set of labeled data (queries and labeled documents), and thereafter used to retrieve relevant documents—all within a single, unified model. As opposed to retrieve-then-rank approaches, this type of model allows for simple end-to-end training and can easily be used as a differentiable sub-component of a larger, more complex neural model.

## 3.1 Indexing Strategies

We investigate various indexing strategies that are meant to learn associations between documents and their identifiers. We train our model to predict docids given a sequence of document tokens. This allows our model to learn which identifier belongs to which document and can be thought of as a differentiable take on traditional search indexes. We consider various alternatives and ablate these settings in subsequent sections. The final strategy employed was Inputs2Targets with direct indexing.

### 3.1.1 Indexing Method

This section discusses the indexing task variants that we consider.

**Inputs2Target**  We frame this as a seq2seq task of `doc_tokens` → `docid`. As its name suggests, this binds the docids to the document tokens in a straightforward inputs-to-targets fashion. The advantage here is that the identifier is the denoising target, which puts it in closer proximity to the loss function. Since the retrieval task is also concerned with predicting identifiers, this formulation allows the network to follow a similar input-target balance in terms of sequence length. A potential weakness is that the document tokens are not denoising targets and therefore there is no opportunity for general pre-training on document tokens.

**Targets2Inputs**  This formulation considers the opposite of the above, i.e., generating document tokens from identifiers, i.e., `docid` → `doc_tokens`. Intuitively, this is equivalent to training an autoregressive language model that is conditioned on the docid.

**Bidirectional**  This formulation trains both Inputs2Targets and Targets2Inputs within the same co-training setup. A prefix token is prepended to allow the model to know which direction the task is being performed in.

**Span Corruption**  We also explored a setup that performs span corruption-based denoising (Raffel et al., 2019) with the inclusion of docid tokens. In this approach, we concatenate the identifier to the document tokens as a prefix that can be randomly masked as spans in the span corruption objective.

This method has the advantage of (1) also performing general pre-training during indexing and (2) achieving a good balance of docids as denoising targets and inputs.

### 3.1.2 Document Representation Strategies

In the previous section, we explored *"how to index"*. This section investigates *"what to index?"*, i.e., how to best represent *doc_tokens*. We state our options here and carefully ablate them in our experiments later. The best option in the end was the direct indexing method.

**Direct Indexing**  This strategy represents a document exactly. We take the first $L$ tokens of a document, with sequential order preserved, and associate them with the docid.

**Set Indexing**  Documents may contain repeated terms and/or non-informative words (e.g., stop-words). This strategy de-duplicates repeated terms using the default Python `set` operation and removes stopwords from the document. The rest of the document after filtering is passed into the model in similar fashion to the direct index.

**Inverted Index**  This strategy maps chunked documents (contiguous blocks of tokens) instead of entire documents directly to the docid. We randomly subsample a single contiguous chunk of $k$ tokens and associate them with the docid. The key advantage of this approach is to allow looking beyond the first $k$ tokens.

### 3.2 Representing Docids for Retrieval

Retrieval within seq2seq-based DSI models is accomplished by decoding docids given an input query. How to do this decoding in an effective way largely depends on how docids are represented in the model. The remainder of this section explores a number of possible ways for representing docids and how to handle decoding for each.

**Unstructured Atomic Identifiers**  The most naive way to represent documents is assign each an arbitrary (and possibly random) unique integer identifier. We refer to these as *unstructured atomic identifiers*.

With these identifiers, an obvious decoding formulation is to learn a probability distribution over the identifiers. In this case, models are trained to emit one logit for each unique docid ($|N_{documents}|$). This is analogous to the output layer in standard language models, but extended to include docids.

To accommodate this, we extend the output vocabulary of a standard language model as follows:

$$O = \text{Softmax}([W_{tokens};\ W_{docs}]^T\ h_{last})$$

where $[;]$ is the row-wise concatenation operator, $W_{tokens} \in \mathbb{R}^{d_{model} \times |N_{tokens}|}$ and $W_{docs} \in \mathbb{R}^{d_{model} \times |N_{documents}|}$. $h_{last}$ is the last layer's hidden state ($\in \mathbb{R}^{d_{model}}$) of the decoder stack. To retrieve the top-k documents for a given query, we simply sort the output logits and return the corresponding indices. This is also reminiscent of standard listwise learning to rank where all documents are considered at once.

**Naively Structured String Identifiers**  We also consider an ostensibly absurd approach that treats unstructured identifiers, i.e., arbitrary unique integers, as *tokenizable* strings. We refer to these as *naively structured identifiers*.

In this formulation, retrieval is accomplished by decoding a docid string sequentially one token at a time. This eliminates the need for the large softmax output space that comes with unstructured atomic identifiers. It also eliminates the need to learn embeddings for each individual docid.

When decoding, beam search is used to obtain the predicted best docid. With this strategy, it is less straightforward to obtain a top-k ranking. One could exhaustively comb through the entire docid space and obtain the likelihood of each docid given the query. Instead, we use the partial beam search tree to construct top-k retrieval scores. We find this approximation to be quite efficient and effective in practice.

**Semantically Structured Identifiers** All of the approaches for representing docids thus far assumed that the identifiers are assigned in an arbitrary manner. While exploring the limits of arbitrary identifiers is quite interesting, it is only intuitive that imbuing the docid space with semantic structure can lead to better indexing and retrieval capabilities. As such, this section explores *semantically structured identifiers*.

Specifically, we aim to automatically create identifiers that satisfy the following properties: (1) the docid should capture some information about the semantics of its associated document, (2) the docid should be structured in a way that the search space is effectively reduced after each decoding step. This results in identifiers where semantically similar documents share identifier prefixes.

In this work, we treat this as a fully unsupervised pre-processing step. However, as part of future work it may be possible to integrate and automatically learn semantic identifiers in a fully end-to-end manner.

To construct identifiers with this property, we employ a simple hierarchical clustering process over document embeddings to induce a decimal tree (or more generally, a trie).

Given a corpus to be indexed, all documents are clustered into 10 clusters. Each document is assigned an identifier with the number of their cluster from 0-9. For every cluster containing more than $c$ documents, the algorithm is applied recursively, with the next level's result (the remaining suffix of the identifier) appended to the existing identifier.

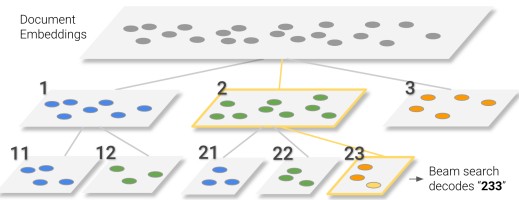

Figure 2: Visual example of a hierarchical clustering process used to assign semantically structured identifiers. During inference, beam search navigates this trie to decode the correct docid.

---

**Algorithm 1** Generating semantically structured identifiers. (Referenced in Section 3.2.)

---

**Input:** Document embeddings $X_{1:N}$, where $X_i \in \mathbb{R}^d$
**Output:** Corresponding docid strings $J_{1:N}$
**function** GENERATESEMANTICIDS($X_{1:N}$)
  $C_{1:10} \leftarrow Cluster(X_{1:N},\ k = 10)$
  $J \leftarrow$ empty list
  **for** $i = 0$ **to** 9 **do**
    $J_{current} \leftarrow [i] * |C_{i+1}|$
    **if** $|C_{i+1}| > c$ **then**
      $J_{rest} \leftarrow$ GENERATESEMANTICIDS($C_{i+1}$)
    **else**
      $J_{rest} \leftarrow [0, \dots, |C_{i+1}| - 1]$
    **end if**
    $J_{cluster} \leftarrow$ elementwiseStrConcat($J_{current}, J_{rest}$)
    $J \leftarrow J$.appendElements($J_{cluster}$)
  **end for**
  $J \leftarrow$ reorderToOriginal($J,\ X_{1:N},\ C_{1:10}$)
  **return** $J$
**end function**

---

For clusters with $c$ documents or less, each element is assigned an arbitrary number from 0 to at most $c - 1$ and likewise its digits are appended to the existing identifier. Although this specific process induces a decimal tree, it is possible to induce similar types of tries using any number of other reasonable strategies. In practice, we simply apply $k$-means over embeddings generated by a small 8-layer BERT model, with $c = 100$. We include pseudo-code for this process in Algorithm 1.

## 3.3 Training and Optimization

The DSI models that we train are optimized for seq2seq cross entropy loss and are trained with teacher forcing. We explored two main strategies for training DSI models. The first and more straightforward strategy is to first train a model to perform indexing (memorization), followed by a fine-tuning stage where the trained model is used to map queries to docids (e.g., retrieval). The second strategy is to train them together in a multi-task setup. To this end, we frame co-training tasks in similar fashion to T5-style co-training (e.g., using task prompts to differentiate them). The latter performed significantly better, especially when the proportion of indexing to retrieval task examples is high. Hence, we adopted multi-task learning as the default strategy.

Here, we make the observation that our setup is unique and unlike traditional multi-task learning or transfer learning. In typical multi-task setups, two tasks have shared commonalities that could improve the performance of both tasks if they were learned together. However, in our setup, the retrieval task is *completely dependent* on the indexing task. In particular, without the indexing task, the identifiers leveraged by the retrieval task would be completely meaningless. Hence, in order to solve task B (retrieval), the model needs to learn task A (indexing) well enough. This problem

setup presents unique and largely unexplored research challenges that might be of interest to the ML community.

# 4 Experiments

In this section, we discuss our experimental setup, datasets used and baselines compared. We also discuss experimental results, findings and effect of various strategies discussed in earlier sections of the paper. Since this is fairly new concept, this work aims to put forth a *proof-of-concept* and seeks to answer research questions instead of making a *'sotaeesque'* comparison. We leave extensive comparisons on other setups and baselines to future work.

**Dataset**  We conduct our experiments on the challenging Natural Questions (NQ) (Kwiatkowski et al., 2019) dataset. NQ consists of 307K query-document training pairs and 8K validation pairs, where the queries are natural language questions and the documents are Wikipedia articles. Given a question, the retrieval task is to identify the Wikipedia article that answers it. For evaluating how DSI models perform at different scales, we construct three sets from NQ to form our testbed, namely NQ10K, NQ100K, and NQ320K denoting different numbers of total query-document pairs in the combined train and validation splits. NQ320K is the full NQ set and uses its predetermined training and validation split for evaluation purposes. Unlike NQ320K, NQ10K and NQ100K constructs randomly sampled validation sets. For all datasets, we use the same docid space/budget of 320K tokens for all unstructured atomic and naively structured identifier experiments. Semantically structured identifiers are generated separately for each dataset so as to prevent leakage of semantic information from larger splits into smaller ones. Text is lowercased. Note that there exists fewer unique documents than query-document pairs in these datasets. Please refer to Table 4 (Appendix) which reports the statistics of these datasets.

**Metrics**  We evaluate our models on Hits@N where N=$\{1, 10\}$. This metric reports the proportion of correct documents ranked in the top $N$ predictions.

**Implementation Details**  All DSI models are initialized using standard pretrained T5 (Raffel et al., 2019) model configurations. The configurations names and corresponding number of model parameters are: Base (0.2B), Large (0.8B), XL (3B) and XXL (11B). For unstructured atomic identifiers runs, we initialize the identifiers randomly as new parameters and only finetune the weights during the indexing stage. We use the Jax/T5X [2] implementation for our experiments. The DSI models are trained for a maximum of 1M steps using a batch size of 128. We pick the best checkpoint based on retrieval validation performance. Our training hardware consists of 128-256 TPUv4 chips for models above 1B parameters and 64-128 TPUv3 or TPUv4 chips otherwise. As an estimate, models above 1B parameters typically take about at least a full day for convergence for NQ320K. We tune the learning rate amongst $\{0.001, 0.0005\}$ and linear warmup amongst $\{10K, 100K, 200K, 300K\}$ and/or none. Semantically structured identifiers are generated using an 8-layer BERT (Devlin et al., 2018) model [3], and the default $k$-means clustering in scikit-learn. Based on our early ablation experiments of various DSI setting, the main results presented use direct indexing ($L = 32$) and the `Inputs2Targets` indexing strategy. We present results for all the docid representation methods. Following the main results, we present our ablation studies.

## 4.1 Baselines

For baselines, we use T5-based dual encoders implemented by (Ni et al., 2021). We use the gensim[4] package for computing BM25 scores. For the T5-based dual encoders, we train with contrastive learning on the NQ pairs until convergence ($\approx 10K$ steps) and obtain top-k nearest neighbors with a system similar to ScaNN (Guo et al., 2020). For zero-shot retrieval, we also compare with a state-of-the-art unsupervised baseline, Sentence T5 (Ni et al., 2021) which have been specially pre-trained with a similarity learning task. There two reasons why we consider (Ni et al., 2021) the relevant dual encoder baseline for this work rather than other dense retrieval works such as DPR (Karpukhin et al.,

---

[2] `https://github.com/google-research/t5x`
[3] `https://tfhub.dev/google/collections/bert`
[4] `https://pypi.org/project/gensim`

Table 2: Experimental results on NQ document retrieval. DSI outperforms BM25 and Dual Encoder baselines. Among all the Docid representation methods, Semantic String Docids perform the best.

| Model | Size | Params | Method | NQ10K | | NQ100K | | NQ320K | |
|---|---|---|---|---|---|---|---|---|---|
| | | | | Hits@1 | Hits@10 | Hits@1 | Hits@10 | Hits@1 | Hits@10 |
| BM25 | - | - | - | 12.4 | 33.5 | 20.9 | 46.4 | 11.6 | 34.4 |
| T5 | Base | 220M | Dual Encoder | 16.2 | 48.6 | 18.7 | 55.2 | 20.5 | 58.3 |
| T5 | Large | 800M | Dual Encoder | 18.8 | 55.7 | 22.3 | 60.5 | 22.4 | 63.3 |
| T5 | XL | 3B | Dual Encoder | 20.8 | 59.6 | 23.3 | 63.2 | 23.9 | 65.8 |
| T5 | XXL | 11B | Dual Encoder | 22.1 | 61.6 | 24.1 | 64.5 | 24.3 | 67.3 |
| DSI | Base | 250M | Atomic Docid | 13.0 | 38.4 | 23.8 | 58.6 | 20.7 | 40.9 |
| DSI | Large | 800M | Atomic Docid | 31.3 | 59.4 | 17.1 | 52.3 | 11.6 | 37.6 |
| DSI | XL | 3B | Atomic Docid | 40.1 | 76.9 | 19.0 | 55.3 | 28.1 | 61.9 |
| DSI | XXL | 11B | Atomic Docid | 39.4 | 77.0 | 25.3 | **67.9** | 24.0 | 55.1 |
| DSI | Base | 250M | Naive String Docid | 28.1 | 48.0 | 18.7 | 44.6 | 6.7 | 21.0 |
| DSI | Large | 800M | Naive String Docid | 34.7 | 60.5 | 21.2 | 50.7 | 13.3 | 33.6 |
| DSI | XL | 3B | Naive String Docid | 44.7 | 66.4 | 24.0 | 55.1 | 16.7 | 58.1 |
| DSI | XXL | 11B | Naive String Docid | 46.7 | **77.9** | **27.5** | 62.4 | 23.8 | 55.9 |
| DSI | Base | 250M | Semantic String Docid | 33.9 | 57.3 | 19.0 | 44.9 | 27.4 | 56.6 |
| DSI | Large | 800M | Semantic String Docid | 37.5 | 65.1 | 20.4 | 50.2 | 35.6 | 62.6 |
| DSI | XL | 3B | Semantic String Docid | 41.9 | 67.1 | 22.4 | 52.2 | 39.1 | 66.8 |
| DSI | XXL | 11B | Semantic String Docid | **48.5** | 72.1 | 26.9 | 59.5 | **40.4** | **70.3** |

Table 3: Experimental results on Zero-Shot NQ document retrieval. DSI outperforms BM25, T5 embeddings and SentenceT5, the state-of-the-art for unsupervised similarity modeling. Among Docid representation method, the Atomic Docid performs the best on zero-shot learning.

| Model | Size | Method | NQ10K | | NQ100K | | NQ320K | |
|---|---|---|---|---|---|---|---|---|
| | | | Hits@1 | Hits@10 | Hits@1 | Hits@10 | Hits@1 | Hits@10 |
| BM25 | - | - | 12.4 | 33.5 | 20.9 | 46.4 | 11.6 | 34.4 |
| T5 | XXL | Dual Encoder | 0.3 | 1.3 | 1.9 | 8.0 | 1.1 | 5.9 |
| SentenceT5 | Large | Dual Encoder | 17.6 | 50.7 | 17.4 | 50.8 | 16.9 | 51.0 |
| DSI | XXL | Atomic Docid | 25.7 | 60.1 | **23.0** | **57.3** | **25.1** | **56.6** |
| DSI | XXL | Naive String Docid | 43.4 | 67.4 | 17.4 | 41.5 | 9.2 | 22.6 |
| DSI | XXL | Semantic String Docid | **43.9** | **68.8** | 11.4 | 26.6 | 13.9 | 31.1 |

2020). Firstly, we employ the exact identical pretrained model, which allows systematic ablation of the proposed approach without conflating other factors. Scientifically, we believe this comparison against fine-tuned T5 is the best apples to apples comparison that we provide. Secondly, fine-tuned T5 dual encoders are considered to be architecturally and methodologically very identical to DPR (with some minor differences such as parameter sharing but use the same concept of in-batch negatives).

## 4.2 Experimental Results

Table 2 reports retrieval results for NQ10K, NQ100K, and NQ320K with finetuning and Table 3 reports zero-shot retrieval results. For zero-shot retrieval, the model is only trained on the indexing task and not the retrieval task, so the model sees no labeled query → docid data points. Section 7.2 of the Appendix reports extended results regarding the indexing performance and training dynamics of DSI.

**Supervised Finetuning Results** Our results show that DSI outperforms DE across all dataset sizes. On the small dataset (NQ10K), the performance gap between DSI and DE is large, e.g., the best DSI variant outperforms DE by 2 times. On NQ100K, the gap becomes less prominent with the best DSI model (unstructured atomic identifiers) outperforming DE by +5% Hits@1 and Hits@10. On the large dataset (NQ320K), the best DSI model (structured semantic identifiers) outperform the best DE model by +66% relative Hits@1 and +4.5% Hits@10.

**Zero-Shot Results** Table 3 reports results on zeros-shot retrieval. Recall that zero-shot retrieval is performed by only performing indexing and not the retrieval task. In other words, the model does not see any annotated query or document pairs. Generally, the best result is obtained by DSI with

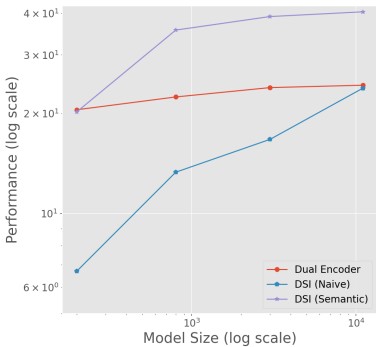

Figure 3: Scaling plots for DSI vs. DE across model sizes. Performance refers to the Hits@1 metric.

Figure 4: Effect of multi-task ratio of indexing to retrieval examples.

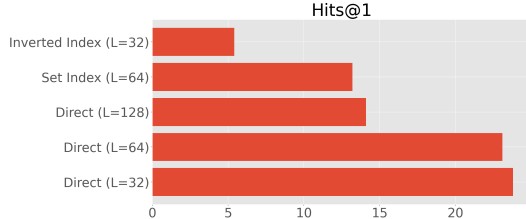

Figure 5: Performance of different document representations. (Referenced in Section 4.2.)

unstructured atomic identifiers on both NQ100K and NQ320K. The best performance on all NQ datasets outperform well-established unsupervised retrieval baselines such as BM25. Moreover, DSI outperforms unsupervised representation learning methods such as SentenceT5 (Ni et al., 2021), which is trained to learn similarity-aware representations via contrastive learning. We also note that raw T5 embeddings perform extremely poorly and do not produce reasonable results on the task of unsupervised retrieval. Given that it is generally difficult for an unsupervised neural method to outperform BM25, we find these early results very encouraging.

**Document Identifiers** One key research question in this paper is the crucial choice of how to represent docids. Generally, we find that structured semantic identifiers are helpful and improve over unstructured identifiers. When comparing naive versus semantic string identifiers, it seems imperative to use semantic identifiers if possible. This is intuitive, since imbuing the target space with semantic structure can facilitate greater ease of optimization and additional unsupervised representation learning methods as external knowledge. The competitiveness of unstructured atomic identifiers is somewhat mixed and we had some difficulty optimizing such models. We hypothesize that this could possibly be because of the the newly initialized softmax layer and that training such a system from scratch would mitigate these issues. However, we defer this line of investigation to future work. In lieu of the instability and high variance of the unstructured atomic identifiers, the performance is not consistent across the different datasets. Moreover, these docids might also run into intermittent non-convergence which we trace back to an optimization related quirk. However, we also note that unstructured atomic identifiers perform the best, by a wide margin, on the zero-shot retrieval setup and achieve performance often more than double than that of beam decoding methods.

**Indexing Strategies** In this section, we explore the effect of different indexing methods (Section 3.1.1). We run experiments on NQ100K with the different indexing strategies described earlier. Models are trained using the Naive Docid method. Without indexing, the model achieves $0\%$ Hits@1. This is intuitive, since the Docids are not meaningful without the indexing task. Secondly, the `Inputs2Targets` and `Bidirectional` formulation performs the best, with the bidirectional method performing slightly worse (13.5 vs 13.2) compared to the former. Finally, the accuracy with Targets2Inputs and Span Corrpution with Docids yield no meaningful results ($0\%$ accuracy). This

goes to show that there can be huge variance across indexing strategies whereby some strategies work reasonably well and some completely do not work at all.

**Document Representations**   In this section, we explore the performance of the different document representation strategies described in Section 3.1.2. Figure 5 reports the results on NQ320K. Overall, we find that the direct indexing approach works the best. We also find that it is difficult to train the inverted index method since the docid is repeatedly exposed to different tokens. We also find that shorter document lengths seem to work well where performance seems to substantially dip beyond $64$ tokens suggesting that it might be harder to optimize or efficiently memorize when there are a larger number of document tokens. Finally, we also find that there was no additional advantage in applying set processing or stopwords preprocessing to the document tokens.

**Scaling Laws**   Another interesting insight is how the scaling law of DSI differs from Dual Encoders. Understanding the scaling behaviour of Transformers have garnered significant interest in recent years (Kaplan et al., 2020; Tay et al., 2021; Abnar et al., 2021). We find that the gain in retrieval performance obtained from increasing model parameterization in DE seems to be relatively small. Conversely, the scaling properties of DSI seems to be more optimistic.

Figure 3 plots the scaling behaviour (log scale) of three methods (DE and DSI with naive and semantic IDs). DSI (naive) strongly benefits from scale going from base to XXL and seems to still have headroom for improvement. Meanwhile, DSI (semantic) starts off equally competitive as DE base but performs much better with scale. DE models, unfortunately are more or less plateaued at smaller parameterization.

**Interplay Between Indexing and Retrieval**   Our early experiments showed that first learning the indexing task and then learning the retrieval task in a sequential manner results in mediocre performance. There, we focused on exploring good ratios $r$ for co-training the indexing and retrieval tasks together using multi-task learning. Figure 4 shows the effect of modifying the ratio of indexing to retrieval samples. We find the optimization process is significantly influenced by the interplay between the indexing and retrieval tasks. Setting $r$ too high or low generally resulted in poor performance. We find that a rate of 32 generally performed well.

## 5   Conclusion

This paper proposed the Differentiable Search Index (DSI), a new paradigm for learning an end-to-end search system in a unified manner, paving the way for next generation search (Metzler et al., 2021). We define novel indexing and retrieval tasks that encode the relationship between terms and docids completely within the parameters of a Transformer model. The paper proposed a number of different ways to represent documents and docids, and explored different model architectures and model training strategies. Experiments conducted on the Natural Questions data set show that DSI performs favorably against common baselines such as BM25 and dual encoders, both in a standard fine-tuning setup as well as in a zero-shot setup.

Although the models and results presented here are promising, there is a great deal of potential future research that can be explored based on this work to improve this approach. For example, it would be interesting to explore alternative strategies for representing documents and docids, as well as to investigate mixture-of-expert models (Du et al., 2021; Fedus et al., 2021; Lepikhin et al., 2020) for scaling the memory capacity of DSI. One important direction will also be to explore how such models can be updated for dynamic corpora, where documents may be added or removed from the system. Finally it may also be interesting to further investigate DSI as an unsupervised representation learning method and/or memory store for other language models to leverage.

## 6   Acknowledgements

The authors would like to thank you Fernando Pereira, Huaixiu Steven Zheng, Sebastian Ruder, Adam D. Lelkes, Ian Wetherbee and Dani Yogatama for their valuable feedback and discussions. We would also like to extend a special thanks to Sanket Vaibhav Mehta for additional experimental contributions.

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
