# 7   Appendix

Here we include figures containing additional details about our experiments.

## 7.1   Dataset Statistics

Table 4: Statistics of NQ datasets used in our experiments. (Referenced in Section 4.) The number in the dataset name corresponds to the total number of document-query pairs in the dataset, while $|D|$ corresponds to the number of unique documents, based on the first 4000 UTF-8 characters of the document.

| Dataset | $|D|$ | Train Pairs | Val Pairs | $V_{doc\_out}$ |
|---------|-------|-------------|-----------|----------------|
| NQ10K   | 10K   | 8K          | 2K        | 320K           |
| NQ100K  | 86K   | 80K         | 20K       | 320K           |
| NQ320K  | 228K  | 307K        | 8K        | 320K           |

## 7.2   Extended Results

We report additional results and observations here.

### 7.2.1   Indexing/Memorization Performance

Table 5: Indexing performance (memorization) on NQ documents via the `Inputs2Targets` indexing objective. All models were indexed on all NQ documents (train and validation), with memorization evaluated on only the documents in the validation set.

| Size  | Params | Method               | Indexing Hits@1 |
|-------|--------|----------------------|-----------------|
| Base  | 250M   | Atomic Docid         | 85.4            |
| Large | 800M   | Atomic Docid         | 84.9            |
| XL    | 3B     | Atomic Docid         | 88.4            |
| XXL   | 11B    | Atomic Docid         | 92.7            |
| Base  | 250M   | Naive String Docid   | 76.3            |
| Large | 800M   | Naive String Docid   | 92.1            |
| XL    | 3B     | Naive String Docid   | 92.2            |
| XXL   | 11B    | Naive String Docid   | 91.9            |
| Base  | 250M   | Semantic String Docid | 87.6           |
| Large | 800M   | Semantic String Docid | 91.5           |
| XL    | 3B     | Semantic String Docid | 92.6           |
| XXL   | 11B    | Semantic String Docid | 92.0           |

We can observe that indexing performance is relatively strong on NQ across methods and model sizes. It is clear though that increasing model size improves indexing performance.

### 7.2.2   Discussion of DSI Training Dynamics

In this paper, all indexing tasks are trained on the union of documents in both the train and validation splits of Natural Questions (NQ). This aligns with traditional definitions of indices, where a document must be in the index in order for the index to retrieve it. Retrieval then is trained on trained only on the NQ train split, with retrieval performance evaluated on NQ validation, based on the best checkpoint.

Analysis following this original work showed that, when training, a DSI model experiences forgetting of previously indexed batches as it indexes new batches, until it loops around again to the next epoch, and processes the same examples again. The indexing task we use in this paper was constructed by concatenating validation documents after the train documents, then applying a buffered shuffle while training the model (sampling the next training batch from a buffer every step). We used a shuffle buffer of size 5000, which is smaller than the size of the validation split for NQ100K and NQ320K.

As a result, DSI experiments in this paper experienced cycles of minimum and maximum *forgetting*, i.e. higher and lower validation scores, based on whether the model had just indexed the validation documents or indexed them one epoch ago, causing regular peaks and valleys in the validation performance. When picking a checkpoint with maximum validation performance, as we do in the main experiments of this paper, we are implicitly then picking the checkpoint with minimum forgetting.

In Table 6, we aim to provide more context to this phenomenon by providing retrieval validation scores for minimum forgetting checkpoints (highest peak), maximum forgetting checkpoints (highest trough), as well as their average score representing if the validation documents were uniformly distributed across the entire indexing split.

Table 6: Additional NQ320K results at minimum forgetting and maximum forgetting checkpoints, with their average (min-forget / max-forget / avg), at Hits@1,5,10,20.

| Size | Params | Method | Hits@1 | Hits@5 | Hits@10 | Hits@20 |
|---|---|---|---|---|---|---|
| Base | 250M | Atomic Docid | 20.7 / 2.6 / 11.7 | 40.2 / 8.6 / 24.4 | 50.9 / 13.0 / 31.9 | 59.2 / 18.8 / 39.0 |
| Large | 800M | Atomic Docid | 11.6 / 2.5 / 7.0 | 30.5 / 7.2 / 18.9 | 37.6 / 10.9 / 24.2 | 46.7 / 15.9 / 31.3 |
| XL | 3B | Atomic Docid | 28.1 / 2.7 / 15.4 | 52.7 / 7.2 / 30.0 | 61.9 / 10.4 / 36.1 | 69.2 / 14.4 / 41.8 |
| XXL | 11B | Atomic Docid | 24.0 / 4.5 / 14.2 | 46.7 / 11.9 / 29.3 | 55.1 / 17.3 / 36.2 | 62.8 / 23.6 / 43.2 |
| Base | 250M | Naive String Docid | 6.7 / 1.5 / 4.1 | 12.6 / 4.3 / 8.4 | 21.0 / 6.0 / 13.5 | 25.6 / 8.1 / 16.9 |
| Large | 800M | Naive String Docid | 13.3 / 2.6 / 8.0 | 26.0 / 7.9 / 16.9 | 33.6 / 11.0 / 22.3 | 40.4 / 14.7 / 27.5 |
| XL | 3B | Naive String Docid | 16.7 / 1.2 / 8.9 | 32.8 / 3.1 / 17.9 | 58.1 / 4.1 / 31.1 | 62.5 / 5.6 / 34.0 |
| XXL | 11B | Naive String Docid | 23.8 / 1.3 / 12.6 | 46.3 / 3.2 / 24.8 | 55.9 / 5.9 / 30.9 | 62.2 / 8.0 / 35.1 |
| Base | 250M | Semantic String Docid | 27.4 / 12.0 / 19.7 | 47.8 / 25.4 / 36.6 | 56.6 / 30.6 / 43.6 | 61.3 / 34.9 / 48.1 |
| Large | 800M | Semantic String Docid | 35.6 / 10.2 / 22.9 | 54.3 / 21.6 / 38.0 | 62.6 / 24.5 / 43.5 | 67.3 / 27.8 / 47.5 |
| XL | 3B | Semantic String Docid | 39.1 / 10.6 / 24.9 | 60.2 / 22.8 / 41.5 | 66.8 / 27.3 / 47.0 | 71.3 / 31.2 / 51.2 |
| XXL | 11B | Semantic String Docid | 40.4 / 12.2 / 26.3 | 60.3 / 24.9 / 42.6 | 70.3 / 30.1 / 50.2 | 74.8 / 35.0 / 54.9 |

*Results.* We see that for the best configuration of DSI (semantic docids), even when experiencing maximum forgetting DSI is still competitive with BM25, and in the average case DSI still outperforms the Dual Encoder baseline.