# OpenReview forum: "Transformer Memory as a Differentiable Search Index"
_NeurIPS.cc/2022/Conference — NeurIPS 2022 Accept_

### Official Review · Reviewer_dTow · 2022-07-04

**Rating:** 6
**Confidence:** 4
**Soundness:** 3 good
**Presentation:** 4 excellent
**Contribution:** 3 good

**Summary:**

The authors propose a new information retrieval framework called DSI that encodes the corpus into the parameters of the model and generates the doc-ids for the queries. In addition, this paper studies variation in how documents and their identifiers are represented, variations in training procedures, and the interplay between models and corpus sizes. Experiments show that DSI outperforms strong baselines such as dual encoder models and has strong generalization capabilities.

**Questions:**

(1)	If the corpus is encoded with the model, it becomes difficult to add or delete documents in the corpus. How to solve this problem?

(2)	For a large-scale corpus, the parameter size of the model is required to be larger. Will this increase the burden of the online IR system?

(3)	Can the process of using doc-id to determine the document itself also be understood as a kind of index? How can this process be distinguished from the definition of the index in IR?


**Limitations:**

The limitations of this work have been described in detail. I think how to dynamically update the documents in the corpus and how to deal with large-scale corpus are issues that authors need to consider.

**Strengths And Weaknesses:**

Strengths:

(1) Propose a novel text-to-text information retrieval paradigm, which has a certain potential for improving information retrieval performance, especially generalization ability.

(2) The experiments are comprehensive and some key variants are discussed.

Weaknesses:

(1) The practical utility of this method needs to be discussed in detail because encoding a large corpus requires a model with large parameters, which makes the model more resource-intensive online.

(2) Using the model to encode the entire corpus makes the interpretability of the IR system worse, and the output becomes more uncontrollable.

(3) In addition, using the model as an index makes it more difficult to add and remove documents from the corpus. For example, a new doc-id is hard to learn by the model if not trained from scratch.

---

> ### Author Response · Authors · 2022-08-02
> **Thank You**
>
> We thank the reviewer for your helpful comments. We respond to your specific questions below, please see the overall comment for our general response.
>
> > Can the process of using doc-id to determine the document itself also be understood as a kind of index? How can this process be distinguished from the definition of the index in IR?
>
> In IR, an index is typically any data structure that aids the efficient retrieval over a corpus of documents. In this case, our learned model is the index. If one desires the original doc content from the id, then yes an additional lookup/join would have to occur. However, this is similar to the existing dual encoder paradigm where some mapping between embedding to document content also has to exist.

---

### Official Review · Reviewer_vdLK · 2022-07-11

**Rating:** 6
**Confidence:** 4
**Soundness:** 4 excellent
**Presentation:** 3 good
**Contribution:** 3 good

**Summary:**

This paper proposes a sequence-to-sequence architecture that predicts the target document ID directly from the query to replace information retrieval approaches that are based on pipeline retrieve-then-rank strategies. The main idea is to build a differentiable search index where all the information about the documents is stored in the parameters of the transformer language model. To achieve that the authors experiment with different document and document ID representations. Evaluation is performed on corpus sizes of increasing size up to 320K and that it outperforms competitive B25 and dual encoder baselines. Notably the results improve when increasing the model size.

**Questions:**

Could the authors summarize which design choices performed poorly in the introduction? Specifically, in lines 50-52 and 70-71. The reader needs to go through model section or reach the last section to figure out what did not work.

What is the retrieval speed of the proposed system compared to existing alternatives e.g. dense retrieval or BM25? Indexing using BM25 should be really fast and the same holds for dense retrieval, especially with MIPS. The trained model on the other hand should generally be much slower to generate the document ID given large pretrained models are known to be slow during decoding. These potential limitations should be highlighted in the text.

Beyond performance on the retrieval tasks, have the authors looked into what extent the documents are being correctly memorized by the pretrained transformer model? For instance, looking at the predicted document ID for a sample of them when part or all the document is given as input to the decoder. It would be interesting to find out if there are any documents that the mode confuses with each other.

Any insights into why T5 XXL performed that poorly in the zero-shot setting? It is severely under-performing across the board compared to SentenceT5.


**Limitations:**

I'd suggest the authors to discuss the following:
- More detailed description of limitations compared to dense retrieval and BM25: e.g. in terms of training difficulty, cost of maintaining/hosting, retrieval speed.
- Potential negative societal impact from memorizing data with pretrained models in an explicit way. Such functionality could indeed have unwanted privacy/confidentiality consequences that deserve some careful thought and discussion.

**Strengths And Weaknesses:**

**Strengths**
- The paper is generally well written and proposes a novel differentiable search index paradigm that uses a sequence-to-sequence transformer model to memorize documents by their IDs and then use it to perform information retrieval tasks.
- Evaluation on datasets of increasing size demonstrate consistent improvements over both dense retrieval and BM25 baselines albeit different variations worked best on different datasets. It is noteworthy that even simple atomic document IDs perform very competitively.
- In dense information retrieval, it is generally very challenging to beat standard baselines such as BM25  and in some cases retrieval focused pre-training is required on multiple datasets (e.g. [1]); let alone for zero-shot retrieval tasks.
- The contributions of this paper open up new avenues for research and should be of interest to both machine learning (memorizing and associative memory) and information retrieval communities (new paradigm for dense information retrieval).

**Weaknesses**
- In terms of experiments, there is no discussion of efficiency limitations of the proposed approach and the ablation studies do not appear to be very thorough; it's unclear how much effort was put into tuning each option.

- Most of the experiments are sufficiently described but there is no code shared in the supplemental material or statement in the paper about code release which could be an issue for reproducibility.

[1] Unsupervised Dense Information Retrieval with Contrastive Learning: https://arxiv.org/pdf/2112.09118.pdf

---

> ### Author Response · Authors · 2022-08-02
> **Thank You**
>
> We thank the reviewer for your helpful comments. We respond to your specific questions below, please see the overall comment for our general response.
>
> > Could the authors summarize which design choices performed poorly in the introduction?
>
> We are happy to edit the introduction to include this summary for a camera ready version.
>
> > Beyond performance on the retrieval tasks, have the authors looked into what extent the documents are being correctly memorized by the pretrained transformer model?
>
> Yes, we do calculate model performance on a sample of the indexing data for all of our models, and we are happy to add these results to the appendix for camera ready. Generally models show strong memorization, ~80-90% exact match performance on indexing on NQ320K (from Base-XXL sizes).
>
> > Any insights into why T5 XXL performed that poorly in the zero-shot setting? It is severely under-performing across the board compared to SentenceT5.
>
> Out of the box embeddings taken from pooling the vanilla T5 encoder outputs, are not at all calibrated for retrieval. The contrastive learning done by SentenceT5 likely corrects the anisotropy of the embedding space. Please see their paper for further discussion. This is likely not an issue for DSI as we do not directly do similarity comparisons in the raw embedding space.

---

### Official Review · Reviewer_GpFY · 2022-07-11

**Rating:** 7
**Confidence:** 4
**Soundness:** 3 good
**Presentation:** 4 excellent
**Contribution:** 3 good

**Summary:**

This paper proposes a sequence-to-sequence style approach to information retrieval, where a transformer-based seq2seq model maps a query (or a document) to a document identifier. The authors consider several approaches to representing both the input documents and the document identifiers, and find that standard document encoding but a “semantically structured” document identifier representation work best. Variants of the proposed model outperform reasonable baselines on Natural Questions-based benchmarks in both fine-tuned and zero shot settings.

**Questions:**

See the last point above for two questions. In addition:
- Do you have any insights into what explains the difference in performance between the various settings, both within Table 2 and between Tables 2 and 3?
- Do you think the semantic string docid approach would work with unseen documents?
- Extremely minor: Table 4 in the appendix makes it seem like NQ320K has only 228K documents; are these just the documents seen at training time perhaps?

**Limitations:**

I think the authors may want to address a bit more explicitly how this approach might scale to documents unseen at training time. Otherwise I think limitations are reasonably well addressed.

**Strengths And Weaknesses:**

Strengths:
- The paper presents a very interesting and provocative idea.
- The methods proposed are simple and reasonable (in a good way!).
- All in all, the paper obtains good performance on the tasks it considers.
- The paper is written very clearly.
- The authors present results for a relatively large number of data-size/model-size configurations.

Weaknesses:
- I think one weakness is that it’s a little bit difficult to understand why the experimental results are what they are. In particular, different model variants (in Table 2) appear to be best in different settings, and the results don’t seem to be monotonic either in the dataset size nor even in the model size in certain cases. Part of the issue here is that the experiments don’t really control for a number of important factors. First, the datasets differ not just in how many total documents they include but also (from Table 4 in the appendix) in the number of training examples. So it’s difficult to ascertain which (if either) is responsible for the change in performance. Similarly, it appears that in the Atomic Docid setting the authors do exact inference, whereas some beam approximation is used for the other two settings. So again it’s difficult to tell if this is a factor too.
- More minor: I think a natural concern about this approach is what happens with documents that have not been seen at training time, and I’d expect the authors to at least discuss this issue.
- Finally, and most minor: the authors view training the model to associate a doc id with a document as a form of differentiable indexing. But is there evidence that this can be used for anything? For example, after training, can we feed only a part of the document to the model and reliably retrieve its id? And if not, is it even necessary to train the model for indexing? Perhaps we could just train the model to associate doc ids with *queries*, which is how we use the model at test time.

Update after response: thanks for your response; I'm increasing my score. I do continue to think, though, that the paper would be stronger if it addressed the potential confounds of number of documents vs. number of training examples and exact vs. approximate inference.

---

> ### Author Response · Authors · 2022-08-02
> **Thank You**
>
> We thank the reviewer for your helpful comments. We respond to your specific questions below, please see the overall comment for our general response.
>
> > what happens with documents that have not been seen at training time
>
> In our current formulation of DSI, all documents must be seen during the indexing training phase, otherwise there would be no way for the model to be aware of the document content to docID assignment that it needs to know. This is analogous to dual encoder systems which have to compute the embedding of a document and write it into a MIPS index before it can retrieve it, the only difference now is that “adding to the index” now is a differentiable process which uses model training to do so.
>
> >the authors view training the model to associate a doc id with a document as a form of differentiable indexing. But is there evidence that this can be used for anything? For example, after training, can we feed only a part of the document to the model and reliably retrieve its id?
>
> Yes, mainly we observe this ability in our Zero-Shot results, whereby after training on *only* indexing, our model is able to retrieve ids given queries which do not necessarily exactly match the original document content.
>
> >Do you have any insights into what explains the difference in performance between the various settings, both within Table 2 and between Tables 2 and 3?
>
> Our intuition for Table 2 is that the problem of DSI does get more challenging with more documents, and having a Transformer with more capacity can help the problem. Further, semantic ids seem to be easier for the model to decode than arbitrary identifiers, as it relies on semantic information which can be associated from the contents of docs and queries. In Table 3, we observe that with just indexing (no queries finetuning) the model is still able to do retrieval for unseen queries.
>
> >Do you think the semantic string docid approach would work with unseen documents?
>
> Under our current design, all documents must be seen during indexing in order to learn the id to document association. However, it may be possible to design an identifier scheme deterministically derivable from the content independently from other documents, e.g. a “fingerprint”. In this case we could potentially consider zero-shot indexing. This is an interesting direction and would be interesting to explore in future work.
>
> >Extremely minor: Table 4 in the appendix makes it seem like NQ320K has only 228K documents; are these just the documents seen at training time perhaps?
>
> There are ~320K examples in NQ, but corresponding to only 228K unique documents.

---

### Official Review · Reviewer_iAXN · 2022-07-17

**Rating:** 7
**Confidence:** 5
**Soundness:** 3 good
**Presentation:** 4 excellent
**Contribution:** 4 excellent

**Summary:**

The paper proposes a generative approach for document retrieval task. Traditional retrieval systems use a dual encoder to map the query and the document to the same vector space, and then perform nearest neighbor search to find the closest document. In contrast to such systems, the proposed system retrieves the target document by mapping each query to the document id in a generative way. In this case, indexing can be done by memorizing the corpus in the parameters of the language model (similar to associative memory). This mechanism has several inherent benefits, one of them being no need to perform indexing which takes a lot of space and time. The paper explores several different ways to index the corpus and represent the document id. The paper shows that inputs2target indexing (learning to generate the target document id with the document content as the input) and semantic document identifier (via clustering) work the best. The results are comparable or better than the dual encoder baselines in Natural Questions 320k dataset (a subset of NQ 320k) and also show promising results in zero-shot setting where the query-document pairs are never observed.

**Questions:**

Questions
- L201: Doesn't Semantically Structured Identifiers also require offline indexing? Then I think the argument that DSI does not requiring offline indexing is not entirely true.
- It would have been easier for me to compare the model with DPR if Table 2 reports Hits@5 and/or Hits@20. Would you be able to provide them?

Typos
- L29: take -> takes
- L110: \citep -> \citet (there are several other places in the paragraph that the authors make the same mistake)
- L220: space between "forcing." and "We"

**Limitations:**

The paper mentions that what the paper proposes is more of a novel idea and perspective on retrieval than a SOTA-ish contribution. I agree with this.

**Strengths And Weaknesses:**

Strengths
- The proposed method is novel. It is a new perspective to think of document retrieval task as a doc id generation. If this works well, it has the potential of complimenting or even replacing traditional dual encoder retrieval paradigm. So I think the paper has a high impact on the research community.
- Zero-shot performance is quite promising. Beating BM25 with an unsupervised model is not easy.

Weaknesses
- The paper experiments on only one dataset (Natural Questions), and on its subset (320k out of 5 million Wikipedia articles). Given that NQ is a very representative dataset in this field, experimenting on only NQ seems fine, but it is not clear why this was not experimented on the full article set. Even if the performance degrades, I think the paper would have looked better with the full Wikipedia results.
- The dual encoder baselines seem to be weak. The paper uses Sentence T5 (Ni et al., 2021), which is not originally designed for retrieval task. A stronger and more appropriate baseline would have been DPR or something similar, which uses in-batch negative training objective to leverage the benefit of dual encoder training.
- The paper does not experimentally compare with a very relevant previous work, AER (Cao et al., 2020). While I agree that DSI is more generally applicable, it would have been great if AER and DSI were comparable in a similar experimental setup to gauge the advantage and disadvantage of doc id generation with respect to title generation.

---

> ### Author Response · Authors · 2022-08-02
> **Thank You**
>
> We thank the reviewer for your helpful comments. We respond to your specific questions below, please see the overall comment for our general response.
>
> > L201: Doesn't Semantically Structured Identifiers also require offline indexing? \
>
> We consider learning semantic identifiers as a preprocessing step that is not strictly tied to indexing akin to building vocabs, pretraining corpora etc. Right now we don’t think it is possible to search the semantic identifiers without building an offline index.
>
> > It would have been easier for me to compare the model with DPR if Table 2 reports Hits@5 and/or Hits@20. Would you be able to provide them?
>
> Great suggestion. These additional metrics were logged during our experimentation process. It would take however some time to compile these results to complete a full table. We can promise to add them to the camera ready version.

---

### Author Response · Authors · 2022-08-02
**General Response**

We thank you all reviewers for your helpful comments. We are happy to see that the reviewers generally appreciate the novelty and potential of the new retrieval paradigm that we propose in this work. Many reviews shared particular themes which we respond to here:

- Limitations of DSI, in particular with regards to capacity, document addition/removal, and efficiency. Given that this is a new (and relatively novel) idea, we wanted to demonstrate a proof-of-concept at a modest scale which we believe is interesting enough to spur more interest in these types of new paradigms. All of these aspects of DSI the reviewers raise merit a much deeper and more rigorous study than the scope of this paper allows. We hope that this paper serves as a foundation for a wider array of research investigating the new problems this paper raises: e.g. what are the capacity limits of DSI? How can we train the index to add/remove documents as efficiently as possible? And how do we make DSI efficient to deploy in practice?

- Experimental design. It was our aim in this paper to create a simple, controlled experimental setting to understand the behavior of DSI under various initial scales: from 10-320K documents, and from Base-XXL parameterizations. As this is a new paradigm, it is key to understand performance at these initial settings before exploring larger scale settings, which we hope to pursue in future work.

- Baseline choices. We targeted baselines that controlled as many variables as possible for this work. The main dual encoder results we provide are using fine-tuned T5 encoders that are pre-trained on the exact identical finetuning data as DSI. There are two key advantages of this comparison. Firstly, we employ the exact identical pretrained model, which allows systematic ablation of the proposed approach without conflating other factors. Scientifically, we believe this comparison against fine-tuned T5 is the best apples to apples comparison that we provide. Secondly, fine-tuned T5 dual encoders are considered to be architecturally and methodologically very identical to DPR (with some minor differences such as param sharing but use the same concept of in-batch negatives). Regarding AER (Cao et al., 2020), we believe that leveraging title metadata from Wikipedia pages in the DSI context can be considered an oracle method, and may make apples to apples comparisons tricky. While it is an interesting question whether it is possible to construct page-title like identifiers for the purpose of DSI, we consider this to be a bigger question out of the scope of this paper.

Further, we thank the reviewers for all the revision suggestions, which we will update the draft with in the future.

---

### Meta-Review · Area_Chair_o3U2 · 2022-08-23

**Recommendation:** Accept
**Confidence:** Certain

**Metareview:**

This paper proposes a differentiable search index paradigm that transforms an IR problem to a generation problem of doc IDs. Reviewers generally find the paper novel and interesting. However, a few limitations are also pointed out. For example, the approach does not work with unseen docs.

**Award:**

No

---

### Decision · Program_Chairs · 2022-09-14

Accept